# Land Concentration and Land Grabbing Processes—Evidence from Slovakia

**Lucia Palšová \*, Anna Bandlerová and Zina Machničová**

Department of Law, Faculty of European Studies and Regional Development, Slovak University of Agriculture in Nitra, Tr. A. Hlinku 2, 94976 Nitra, Slovakia; anna.bandlerova@uniag.sk (A.B.); xmachnicova@uniag.sk (Z.M.)
\* Correspondence: lucia.palsova@uniag.sk; Tel.: +421-37-641-5079

**Abstract:** In Slovakia, the large-scale acquisition of agricultural land in combination with land concentration represents a legitimate threat that can lead to land grabbing. Based on the research, two interrelated areas of protection need to be effectively regulated to limit land grabbing: the protection of access to land and the protection of agricultural land. Due to the absence of relevant data analysis regarding this issue, the main aim of the study was to analyse the emergence of land concentration in Slovakia based on historical and cultural factors and to evaluate the current legislative and institutional framework of both aspects of land protection with a possible impact on the successively graduating threat of land grabbing. In particular, analytical methods were used, presenting the data from secondary literature sources, a questionnaire survey, and representatives of the Ministry of Agriculture and Rural Development in Slovakia. The research shows that although the state has adopted the necessary legal framework for the protection of property rights to agricultural land, it is not possible to enforce it, as the institutional framework for its implementation is absent. It is also the state's malfunctioning land protection regulatory mechanism and the absence of indirect action instruments that may be key indicators leading to the processes of industrial agriculture. Therefore, the adoption of legislation limiting agricultural land acquisition is important, but the processes of land grabbing presume the state's complex provision of a regulatory mechanism and adoption of strategic measures aimed at sustainable land quality and food security.

**Keywords:** agricultural land; land concentration; land grabbing; land withdrawal; land ownership





## 1. Introduction

The sustainability of agricultural land quality represents a prerequisite for the fulfilment of the requirements for the minimum quality of an individual´s life. The modus operandi of the exercise of land ownership rights in accordance with Article 17 of The Universal Declaration of Human Rights [1] correlates with the application of social and economic rights of the second generation of human rights of owners and authorised users of land. A broad distribution of land ownership is considered to be a basis for the welfare of local economies and rural communities where access to land is one of the means of participation in the political and economic life of the country [2–4].

Moreover, in recent decades, the concept of land reform has been broadened in recognition of the strategic role of land and agriculture in development [5]. The use of agricultural land as a natural resource has become a strategic interest of various and often colliding entities that, in specific countries, misuse inefficient processes of structural changes in agriculture [5,6] and unclear land ownership rights when obtaining it. This leads to large-scale land acquisition and land concentration within the power of disposal of a small group of entities. Although it was proven that land concentration can have a positive effect [7,8] (i.e., higher production cost-effectiveness, productivity, employment, and good infrastructure), the behavioural tendency of monopoly entities on agricultural land has negative effects subsumed under the term of land grabbing, which must be regarded as negative [6,9,10].

However, no generally accepted definition of land grabbing that fully captures its characteristics [11] has been adopted yet at the international level. The original perception meant mainly transnational land trade transactions, but it now includes any speculative land transactions [12]. The Tirana Declaration [13] provides one definition that determines land grabbing as land acquisition standing for the violation of human rights, not based on free, prior, or informed consent of the affected land users, in disregard of social, economic, and environmental impacts, and not based on transparent contracts and on effective democratic planning. Its typical definitional characteristics include unregulated market land investments with the aim of industrial farming [14] or speculative investments realised through large-scale land acquisition activities through domestic or supranational companies, governments, and individuals [15–17]. The negative impact of the process lies in environmental injustice in the form of serious social, economic, and environmental consequences and the violation of human rights [18–21]. These include an absence of respect to traditional and non-formal land tenure, a lack of impact of local communities on the decision-making process, restrictions on access to resources, and thus an overall imbalance in negotiation power in terms of land agreements [22,23].

The global consequence of the unprecedented increase in the number of supranational land investments in land and large-scale land acquisition [9,24] may be assumed to lead to a dramatic change in the world agrarian environment [25]. The loss of control over land or limited access to land for small-scale farmers cultivating the majority of farmland throughout the world [18,26] will result in the restructuralisation of small agrarian output models into large agricultural production systems based on intense and commercially focused technologies [9,27] and the transformation of local fading environments through monocultures [11]. The real concern behind the development of large-scale investments in farmland is that giving land away to investors and having better access to capital to "develop" implies huge opportunity costs as it will result in a type of farming that will have less powerful poverty-reducing impacts than if access to land and water were improved for the local farming communities [9].

As far as the European Union is concerned, it has continued to overlook the dynamics of the interest in large-scale acquisition of agricultural land. Data provided by Eurostat [28] show that the number of farms decreased by 4.2 million (approximately 25%) in the period of 2005–2016, while 85% of the overall decrease was represented by small-scale farms lying on an area of less than 5 ha at the expense of an increase in the area of land owned by large farms.

Although there are several initiatives and mapping services that analyse and make publicly available land use data regarding land grabbing, both global (Land Matrix Global Map; Earth Observing System) and pan-European (Copernicus Land Monitoring Service) comprehensive EU data analyses are not available [11,29].

However, the European Parliament, in its study, identified the determinants predicting the existence of land grabbing in the EU, which, in addition to complicated land ownership and a cheap import price of agricultural land, include a dysfunctional system of public authorities [30]. (Research can be based on the premise identified in the analysis conducted by the European Economic and Social Committee [27] and the European Parliament [30], which states that land grabbing occurs particularly in countries where there is complicated land ownership, a cheap acquisition price of agricultural land, or a malfunctioning system of state bodies).

As land becomes a commercial medium and a motivation for political problems, economic and power gains, and self-realisation, the institutional system often fails, and land management becomes one of the most corrupt sectors of public administration [31]. The results of an Oxfam study [32] showed that globally increased purchases or speculative investments in land were made in more than three-quarters of countries that were assessed by the Worldwide Governance Indicators (WGI) database as countries with low efficiency of institutional mechanisms. Comparing the conclusions of land grabbing and weak institutional framework dependence with the European database (Figure 1), the conclusions

of the analysis of the European Economic and Social Committee confirm that land grabbing occurs primarily in the countries of Central and Eastern Europe [27].

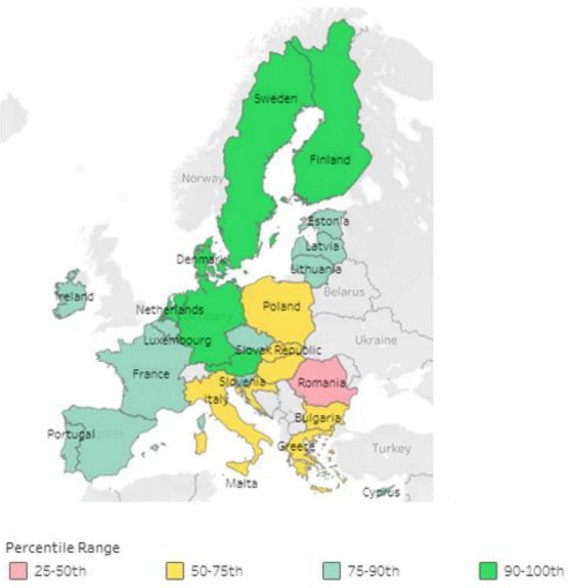

**Figure 1.** Evaluation of governance effectiveness in EU member states according to the WGI database, 2019. Source: Worldwide Governance Indicators—Government Effectiveness [33].

The vulnerability of countries with weak institutional mechanisms caused by the process of land grabbing is supported by the statistical data, indicating that 89.5% of people who worked regularly in agriculture in the EU were the sole holders (farmers) or members of his or her family in 2016; the only Member States where this proportion was lower were Czechia (37.4%) and Slovakia (50.9%) [34]. Comparable results were confirmed in a study by an expert group of the Directorate-General for Internal Policies of the Union [35], stating that Slovakia is facing problems with land use and land grabbing due to a complicated system of land ownership and leasing arrangements that sets Slovakia apart from all other Member States. Land grabbing as a relatively new development phenomenon of land relations in Slovakia is not, as in other countries, a constituted concept of legislation and implementation practice.

The Ministry of Agriculture and Rural Development of the Slovak Republic (MARD) has indirect information about the increased interest in land acquisition by domestic and foreign investors, e.g., Denmark, Italy, China, the USA, and Israel [36], which raises legitimate concerns about land grabbing [37].

In response to this threat, MARD prepared a land reform (amendment to Act No. 140/2014 Coll. on the acquisition of ownership of agricultural land) regulating land grabbing processes, which subsumes under the so-called term "speculativeness" of land acquisition, which is defined with two meanings [36]:

- The purchase of land with the aim of depositing capital while the acquirer is not primarily motivated by the agricultural use of the land;
- The acquisition of land ownership by entities conducting business in agriculture and accumulating their ownership in order to strengthen their competitive position and to take control over the economic chain starting from primary production through processing up to placement in the market.

As deduced from land grabbing research conducted in several countries around the world [10], two interrelated areas of protection need to be effectively regulated to limit land grabbing: the protection of access to land ownership and the protection of the quality and quantity of agricultural land.

In the conditions of Slovakia, both aspects of land protection are significantly determined by the legal and institutional framework for the restructuring of renewed land ownership in the post-socialist period, the examination of which helps us to discern how its prior organisation affects property making and environmental protection at present [38]. In this regard, understanding the intricacies presented by land ownership rights has been a key feature of sustainable development because the terms of access and ownership essentially mediate or structure the relationship that people have with their land, influencing what they do with it and how they treat it [39–41].

As agricultural land is a sui generis subject inherently combining private and public goods (resources), both land protection aspects are often regulated independently, and their imbalance in the legal and institutional framework can lead to the negative effects of the land grabbing process. The aim of the paper is therefore to analyse the emergence of land concentration in Slovakia based on historical and cultural factors and to evaluate the current legislative and institutional framework of both aspects of land protection with a possible impact on the successively graduating threat of land grabbing (Figure 2).

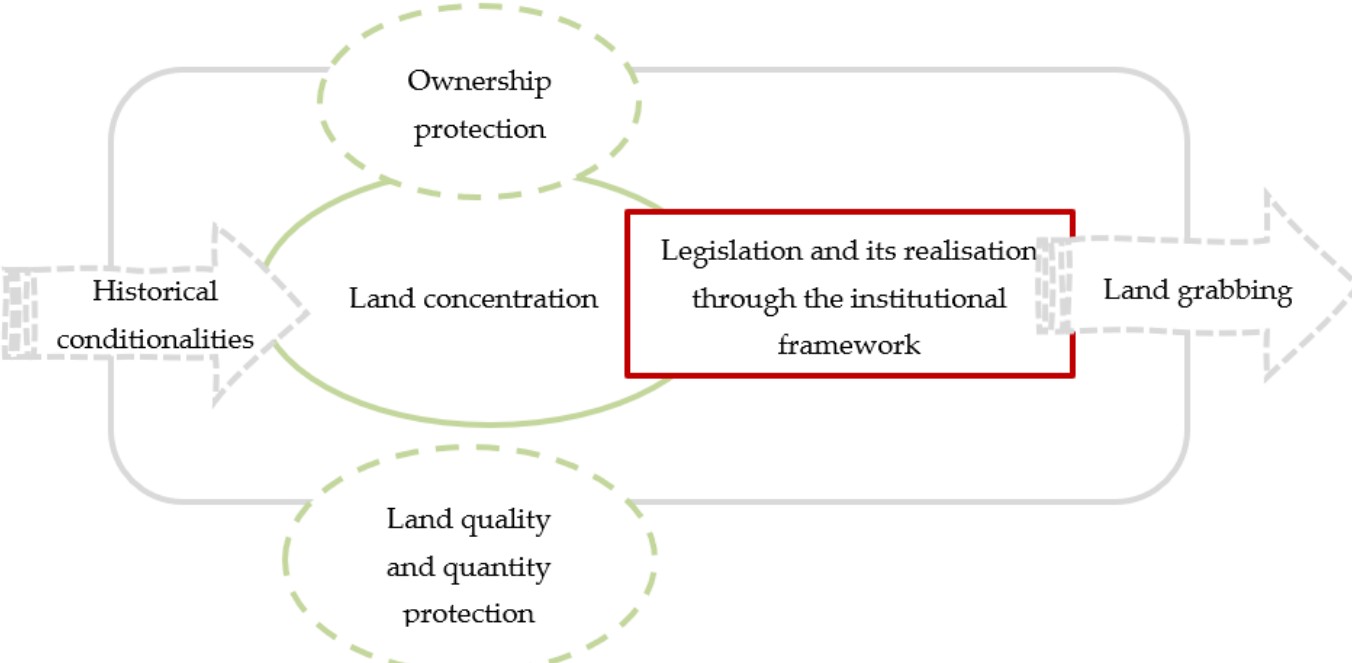

**Figure 2.** Context of the examined connections leading to the threat of land grabbing in Slovakia. Source: own representation, 2021.

Land grabbing is a worldwide-discussed phenomenon which many studies [11,24, 26,30] consider challenging to detect. There is no internationally accepted definition or research framework of this term, which makes it difficult to study.

Also there are no scientific studies in Slovakia concerning the legal and institutional mechanisms for the protection of agricultural land, it is our intention to contribute to scientific discussion about the importance of the examination of this issue with its impact on the efficient management of agrarian pro future policy.

## 2. Materials and Methods

Several studies related to land grabbing have revealed substantial gaps in the existing literature in terms of conceptualisation, methodology, and research area [11,12,37]. Therefore, we created a graphical figure for the purpose of simplifying the presentation of the research methodology (Figure 3).

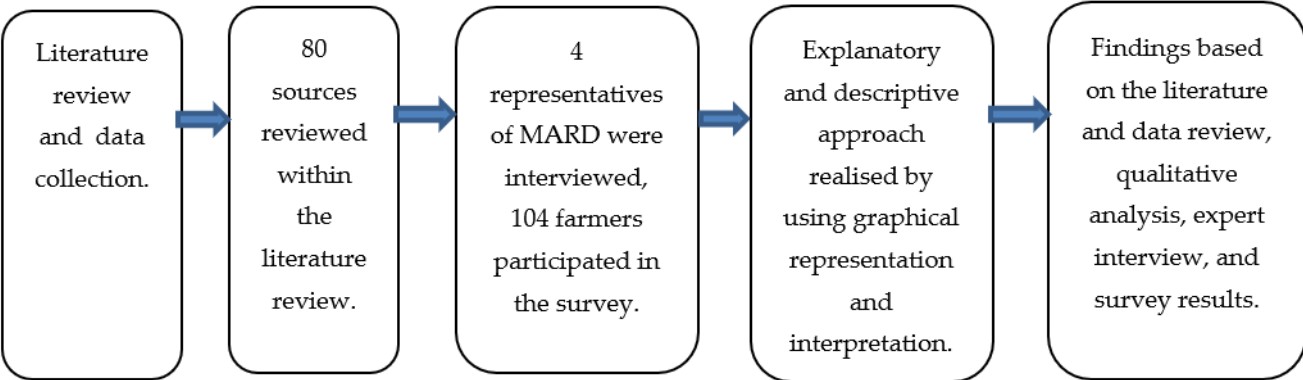

**Figure 3.** Material and methods used in the article. Source: own representation, 2021.

During the research, we analysed data from secondary sources such as scientific studies by researchers in the field, for which we used Google Scholar as a web search tool of scientific literature and its filtering options, focusing on keywords such as land grabbing, land concentration, and industrial agriculture. Other graphically represented and interpreted sources include, in particular, data from the Eurostat database, the Statistical Office of the Slovak Republic, and available data from MARD. The primary source was a guided interview with a MARD representative and a questionnaire survey with Slovak farmers. From a methodological point of view, the paper was developed through qualitative research, as we used methods such as an in-depth analytical approach in the field of theoretical issues, structured interviews, and a questionnaire survey formulated into scientific assumptions, which we subsequently supported or refuted through an analysis of the empirical data. The questionnaire survey consisted of 36 questions (of the open, semi-open, and closed type) focused mainly on the agricultural land cultivation in connection with issues of ownership and legislation. The structured interviews provided insights into these issues from representatives of MARD as the main body of agricultural land protection. We used the methods of analysis, synthesis, and comparison to evaluate the questionnaire survey and structured interviews. The results of these two sources were presented by verbal interpretation.

The data were processed and presented in tables and graphs, based on which we extracted our interpretations. The result was a comprehensive overview of the factors influencing the processes leading to the existence of land grabbing in conditions of Slovakia.

## 3. Results

The results consist mainly of the secondary sources analysis and interpretations supplemented and supported by the results of the questionnaire survey and interviews, which provide a boarder perspective on the situation in Slovakia. As they were part of a wider research project, we used the results as an addition to support our findings.

Slovakia can be characterised as a predominantly rural country, and agriculture remains important in terms of its productive and non-productive functions. Of the total land area in Slovakia (4,903,407 ha), 2,376,712 ha (48.47%) [42] is agricultural land, but it is not of the best quality on average [43] if we consider the population and its demands for the provision of nutrition.

### 3.1. Historical Milestones towards Land Concentration

Historical conditionalities can largely reflect the emergence of land concentration in the conditions of Slovakia. In particular, the period of socialism (1948–1989) was marked by processes of confiscation and collectivisation with the objective of monocultural land management through non-democratically created agricultural companies (cooperatives and state companies).

Ownership as a legal de jure arrangement existed, but all property rights, especially the right and benefits (ius utendi et fruendi), belonged to agricultural companies. Large-scale use of agricultural land by only one entity has gradually become broadly accepted as a holistic agricultural concept of every municipality in Slovakia. Agricultural companies were involved not only in agricultural activities but also in the socio-economic development of the municipality, which provided the citizens with an illusion of rural life quality. According to research [44] based on comparative data from the Maddison Project Database, during socialism, agriculture reached lower productivity and significantly lower living standards in comparison with those in the West at that time. The absence of property rights led to a negative impact on the weakening of responsibility for the land owned, which necessarily led to the loss of economic freedom [45]. All property-related arrangements, in general, were replaced by an administrative system in which the state preferred to control the behaviour of each agent directly, rather than relying on the agent's own pursuit of self-interest [46].

Although the year 1989 brought de jure constitutional renewal of the human and legal concept of property rights to land, as a result of the inconsistent exercise of rights through the institutional framework, the state had not been able to fully implement them in practice.

We can see this result because the state did not sufficiently reflect the necessity to create any legal and institutional support for the construction of property values and responsibility towards the landowners, whom the socialistic system deprived of a set of particular values, personal identity, and emotional bonds [47]. The induced lack of interest of the owners in the realisation of property rights caused the deepening of land fragmentation (legal, technical, and spatial) and the establishment of the legal institute of the so-called unknown or unidentified owner. On the basis of MARD's [36] current data, the following can be identified:

- 8.4 million ownership parcels;
- 4.4 million owners and 100.7 million co-ownership relations;
- An average of 11.93 owners of one parcel;
- 22.73 parcels owned by one owner.

At the same time, the state had purposefully replaced its weak commitment to addressing the restoration of land ownership in the post-socialist period with broad support for the legal framework for the transfer of tenancy and usufruct relations. For this reason, landowners were economically motivated to lease the land rather than take responsibility for their ownership. On the basis of these processes, the state maintained the status quo of large-scale use and continuity of land management of a small group of former agricultural holdings, which transformed their form or transferred the business to new entities [48].

*3.2. Land Concentration in Ownership and Tenure Relationships*

The state's access to ownership of agricultural land influenced the current structure of ownership and usage relations.

As identified from Figure 4, out of the overall area of agricultural land, 77.5% is owned by private entities (individuals and legal entities), 5.77% of land is owned by the state, and 16.77% of land is owned by unknown or unidentified owners administered by the Slovak Land Fund, as a budgetary institution of the state. Therefore, Slovakia belongs to the countries with the highest proportion of state control, in terms of land management. The findings of multiple studies on this point demonstrate that initiatives to control the land (irrespective of whether the object of control is privatisation or nationalisation of land) by the state or a state-like authority often result in unintended negative consequences. One is, in particular, inefficient land management, with non-transparent behaviour of entities influencing the allocation of land management [9–11,32,41,48].

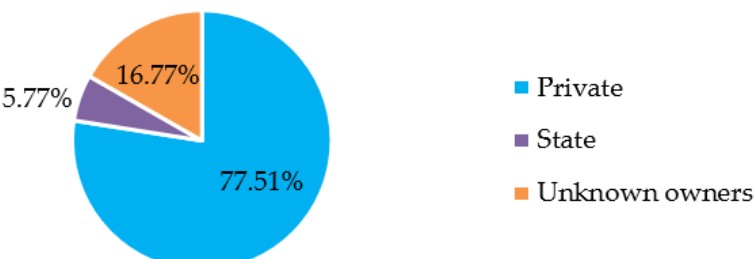

**Figure 4.** Ownership of agricultural land in the year 2018. Source: own representation based on the Annual Report of the Slovak Land Fund, 2019 [49].

When assessing the above-mentioned, restructuralisation of renewed land ownership towards large-scale management in the form of lease relations defines the structure of land management by individual entities (Figure 5). Approximately up to 90% of the total land is leased land [50], which is one of the largest land tenure concentrations in the EU.

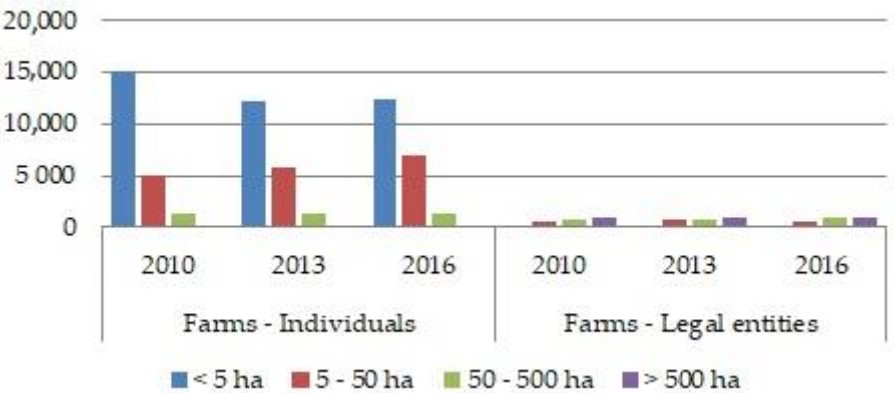

**Figure 5.** Structure of farms according to the size of the cultivated land, 2010–2016. Source: own representation based on data from the Statistical Office of the SR [51–53].

An analysis of land management showed a correlation between the structure of ownership and user relations and the structure of farms according to the size of cultivated land. In particular, individuals tend to manage parcels up to 50 hectares. These occurred in the post-socialistic period and represent entities presenting themselves as individuals—farmers for whom agriculture is a primary or secondary business. In the period observed, the most significant decrease in the number of farmers of land up to 5 ha (decrease of 17.2%) occurred between the years 2010 and 2013. On the contrary, legal entities dominate areas of more than 50 or 100 hectares. Legal entities are represented by their statutory bodies, while several forms of legal entities (stock companies, cooperatives) do not allow publication of the structure of their partners. It seems that another problem is that agricultural land can become any partner´s contribution in kind or a subject of trade and legal relations.

The threat of the creation of dominance in the market [36] motivated the legislator to present a legal obligation for almost all legal entities to register the final user of benefits into the Business Register since 1 November 2018. An analysis of data provided by MARD for 2019 shows that only about 30 final recipients owned agricultural land with an average area of 10 thousand ha, which represents land concentration of 12.6% out of the overall area of agricultural land [49,54]. Despite this, the state did not adopt any other measures to limit or at least monitor land acquisition by individual entities. This approach prevents the state from limiting the land concentration formed by partners of legal entities, which is also criticised by the European Parliament [55]. Several EU countries (e.g., Hungary) either do not allow or limit the acquisition of property rights to land by legal entities.

The need for regulation of the impact of legal entities on land ownership or tenure concentration is demonstrated by the level of land management provided by individuals

and legal entities. In the observed period of 2010–2016, legal entities managed 80% of overall agricultural land. In terms of the size of farms in standard output (Figure 6), large farms, representing 5.2% of all farms, managed 75% of overall agricultural land. On the contrary, very small farms, representing 68.3% of all farms, managed only 3.8% of agricultural land.

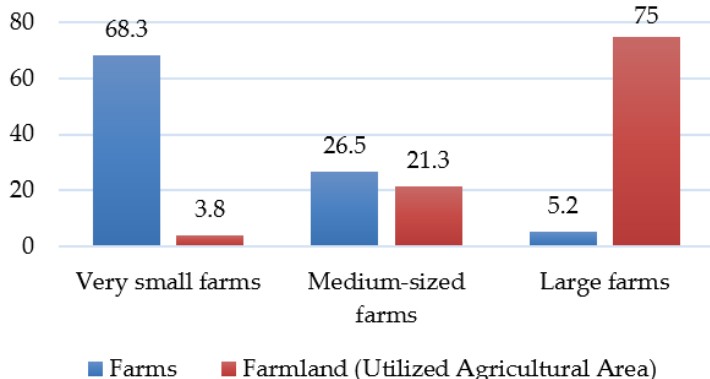

**Figure 6.** Farms and agricultural land in accordance with the size of farms in standard output (SO) as a percentage in 2016. Source: Eurostat (online data code: ef_m_farmleg) [28]. Note: very small farms: SO of <EUR 8000; medium-sized farms: SO of EUR 8000 to EUR 249,999; large farms: SO of >EUR 250,000.

The size structure of farmers´ parcels correlates with the level of monocultural land management. According to data from the Ministry of Environment of the SR [56], in 2020, the average monocultural land parcel situated in Slovakia reached a size of 12 hectares. The average for the EU is 3.9 hectares, and Slovakia has the largest average size of monocultural land parcels among the EU states.

### 3.3. Land Concentration towards Land Grabbing?

The legal and institutional framework of land protection in Slovakia is characterised by the dichotomy of agricultural land protection legislation, with the land being both an object of property rights (private good) and a component of the environment (public good) (Figure 7).

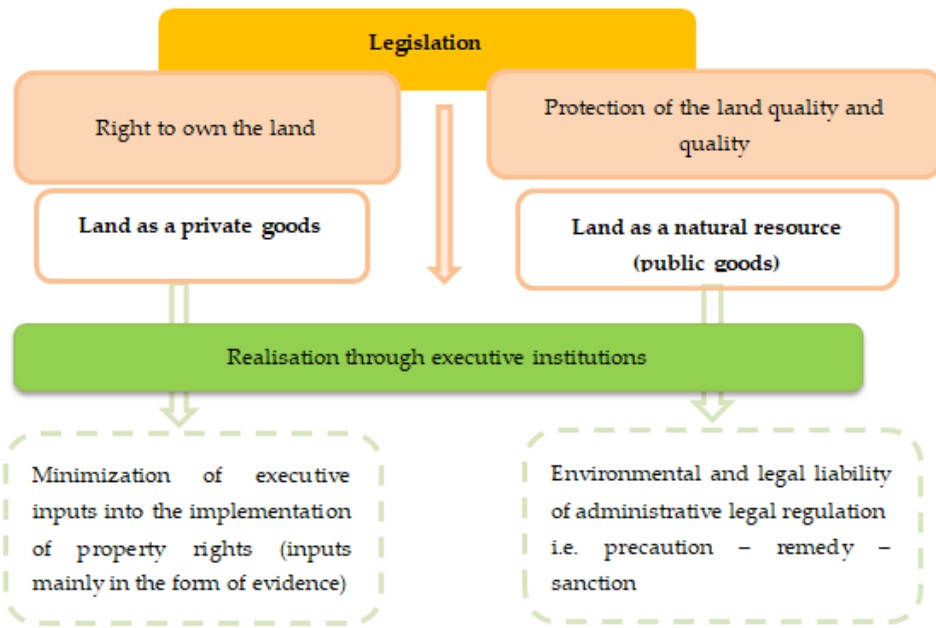

**Figure 7.** Legal regulation of agricultural land in Slovakia. Source: own representation, 2021.

From the point of view of the legal and institutional framework of the protection of agricultural land as an object of property rights, the legislation has a protective character towards the protection of a wide distribution of property rights as a fundamental human right. For this reason, there are also no executive control and accountability mechanisms governing property rights.

Unlimited access to land was the intention for the adoption of the constitutional Act No 137/2017 Coll. amending and supplementing the Constitution of the Slovak Republic [57,58]. The Act defines a possibility of legal interference by the state by means of limitation of property rights to agricultural land in the area of:

- Acquisition of property rights necessary for ensuring public interest, i.e., the needs of society, state food security, development of the national economy, and public interest;
- Legitimate entities in terms of agricultural land acquisition.

Despite the explicit authorisation, the amendment of the Act No 140/2014 Coll. on the acquisition of ownership of agricultural land and on the amendment of certain acts as amended has not been adopted thus far. It suggests the introduction of not only a maximum possible level of agricultural land acquisition by individuals (max. 300 ha) and by legal entities (max. 1200 ha), but also the possibility of the state´s application of its pre-emption right in the case of public interest [36]. Although the Act reflects issues of land grabbing, large-scale farmers reject it because they are disconcerted by the loss of a part of their production tools and difficulties in proving their user rights. At the same time, they state that MARD does not provide sufficient analyses and reasons for the amendment that is being prepared.

On the contrary, the protection of agricultural land as a natural source of public interest is covered by relatively complex legislation. There is general legislation referring to the protection of agricultural land quality offered by the Act No 220/2004 Coll. on the protection and use of agricultural land and amending Act No245/2003 Coll. on integrated prevention and control of environmental pollution and on amendments to certain acts.

The environmental and legal liability is structured in the form of principles of standardised administrative and legal regulation, i.e., precaution–remedy–sanction, which is brought into practice by public authorities at the level of individual legal and application activities [59], which directly embody the protection of agricultural land in specific parcels [60] (Figure 8).

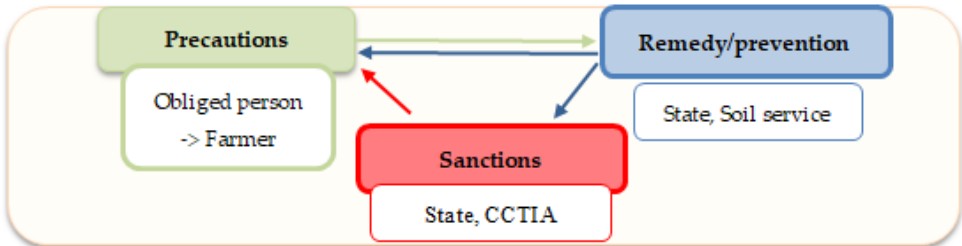

**Figure 8.** Environmental and legal liability for agricultural land protection pursuant to the Act No 220/2004 Coll. Source: own representation based on the Act No 220/2004 Coll. [61].

An analysis of the institutional framework can identify the problem with the implementation of the law in all three areas of legal liability. Assessing individual elements forming environmental and legal liability, it indicates that the protection of agricultural land quality is performed by measures of direct activities [59]. Obligations imposed on the obliged persons are multidisciplinary in terms of knowledge, and failure in their fulfilment results in sanctions.

As the state provides support in the form of an advisory system consisting of methodically uncoordinated advisors [62], the research involved an inquiry among the farmers, and it showed that they gain their knowledge and skills at their own expense in the form of self-study, knowledge exchange, guidance provided by suppliers of protection products,

and vocational training. The absence of economic or optional instruments offered by the state can be assessed as negative, as no stimuli for the persons obliged to protect the agricultural land quality are provided.

The remedy (professional supervision) in terms of agricultural land protection is delivered by the state (agricultural land protection bodies: MARD, District Office in the site of the Self-Government Region and District Office) based on the remedy and prevention activities of Soil Service (Soil Science and Conservation Research Institute, National Agriculture and Food Centre). In order to ensure prevention, remedial bodies are authorised to impose remedial measures to eliminate existing deficiencies with the aim of returning to the initial state (Figure 9), which can be defined as crucial from the perspective of land quality protection.

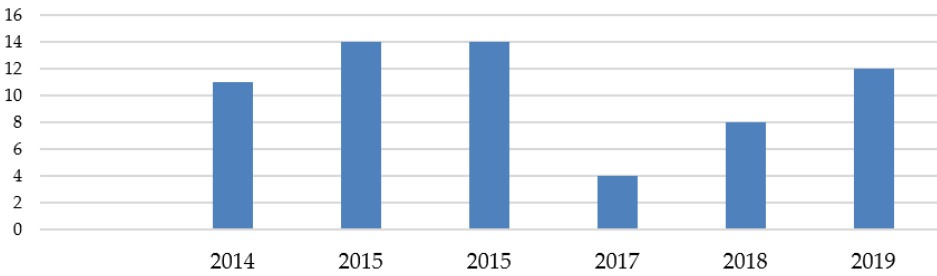

**Figure 9.** Performance of remedy and prevention by the soil service in years 2014–2019. Source: own representation based on the annual report on the activities of the National Agricultural and Food Centre for the years 2015–2020 [63].

Figure 9 shows that the land service performed only a small number of active prevention actions leading to agricultural land protection in the observed period. That may be due to being underfunded. Although the budget for prevention actions has increased gradually (2013: EUR 55,000; 2019: EUR 161,150), it is not possible to ensure the effective application of legal regulations expected by the legislation on such a low budget. This results in the obliged persons being sanctioned for non-compliance with legal regulations without the possibility of a remedy.

Sanctions in the area of land quality are based on the principle of strict liability. Sanctions are imposed by the state through:

- The District Office in the site of the Self-Government Region in the area of general agricultural land quality protection pursuant to the Act No. 220/2004 Coll. (Figure 10);
- The Central Control and Testing Institute in Agriculture in specialised areas (fertilisers, seeds, testing, etc.).

According to Figure 9, in terms of agricultural land protection, there were almost no cases of violations registered in the area of special care for agricultural land in 2014–2019. Several sanctions were imposed in the area of general care, but competent bodies recognised those sanctions as almost exclusively caused by inconsistency in the formal organisation of data related to land quality and the real estate cadastre.

As for the specific areas of land protection, the annual reports of The Central Control and Testing Institute in Agriculture [64] state that, on average, 90% of all imposed sanctions related to formal non-compliance with legal provisions in 2012–2019 (e.g., incorrect evidence of land parcels, unsent reports, etc.).

An analysis of the environmental and legal liability of land protection shows the state´s insufficient implementation of the regulatory mechanism in ensuring compliance with legislation in the area of agricultural land protection. Moreover, we can see the absence of a methodically coordinated advisory system with economic and voluntary instruments based on the state´s indirect application of its influence on the behaviour of regulated entities [48], which would stimulate the obliged persons to implement preventive measures in relation to atmospheric, land, and climatic changes in an adaptable way.

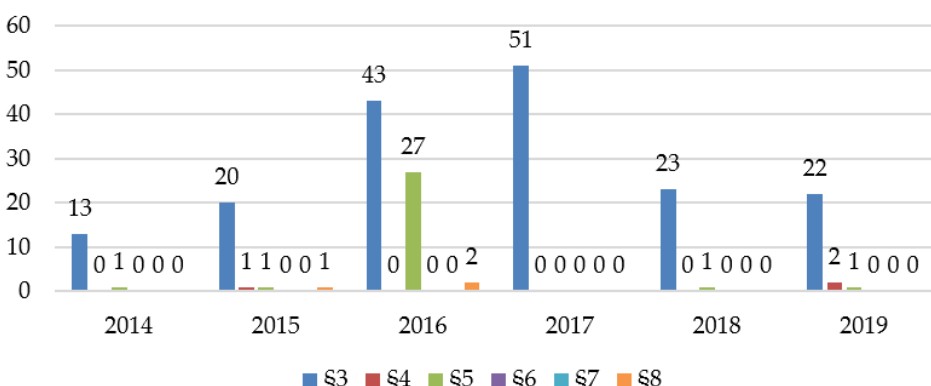

**Figure 10.** Comparison of the number of decisions pursuant to §3–8 Act No 220/2004 Coll. in the years 2015–2019. Source: own representation based on data from MARD [36]. Notes: Pursuant to Act No 220/2004 Coll: §3, general care for agricultural land; §4–§8, special care for agricultural land (§4, protection of agricultural land against degradation; §5, protection of agricultural land from erosion; §6, protection of agricultural land against compaction; §7, preservation of the content and quality of soil organic matter; §8, protection of agricultural land from hazardous substances).

The analysis also shows that the state exercises inadequate and irregular control activities. When those factors are combined with the interest of farmers in increasing their profit, it may result in farmers having a careless approach to compliance with preventive measures. In the long term, such activities may lead to industrial agriculture.

The aforementioned was also confirmed by the Constitutional Court of the SR [60] in the case under sp. zn. PL. ÚS 20/2014, which shows that the state insufficiently implements its responsibility towards real land protection. It also emphasises that the more important the constitutionally protected interest is, the higher the state´s level of responsibility for its efficient protection must be.

## 4. Discussion and Conclusions

Land grabbing and its negative socio-economic and environmental impacts pose a real threat to contemporary agriculture across all EU Member States. Nevertheless, there is currently no unit definition of this term, which complicates the research in this area. Therefore, it is necessary to examine the phenomenon across the individual states in the context of the available scientific literature. Our paper provides an overview of the past and current situation mostly of historical and legislative context of land grabbing in Slovakia as one of the states with the assumption of it´s existence.

Slovakia is more vulnerable because, due to historical conditions and the current complicated access to land, it allows a small group of entities to concentrate the land and to use it as a means of production. Analyses of land ownership in Slovakia show that due to the complicated access, Slovakia has a prevalence of extremely fragmented land.

Such great fragmentation of land ownership causes a situation where it is difficult or impossible to technically register the land. Since registration is a basic precondition for the legal management of land, in individual cases, the land becomes of insignificant economic value [65].

Landowners are not able to withdraw their land plots because they own very small, scattered land plots, and they are often deprived of access to them [66]. Therefore, land fragmentation places the landowners in an unequal position compared to the owners of other commodities, and it causes a disadvantaged position of agriculture compared to other sectors [67].

State failure in relation to the issue of restored ownership and targeted support for leases in the post-socialist period (approximately 90% of all land is leased) has caused extreme concentration of land tenure in the hands of a small group of large farms and legal entities (5.3% of all farmers). Between the years 2010 and 2019, they farmed on 80% of all

agricultural land. There is then a paradoxical phenomenon in which a greater fragmentation of land ownership leads to homogenisation of use and subsequent monocultural land management [68]. Landowners, as the dominant land users, then proceed with lax behaviour [69] and acquire agricultural land at relatively low prices.

These factors, in combination with the current legislation [70], permit the acquisition of ownership of agricultural land in principle without any restrictions. They also enable the dominant position of farmers in the market and allow industrial agriculture to be a form of land grabbing. Its characteristic feature is the abuse of the dominant position of farmers in the market, which leads to a reduction in the prices of agricultural commodities by producing large quantities of cheap food at the expense of environmental damage (degradation of groundwater, surface water, soils, and biodiversity) and displacement of small farmers [71,72].

In Slovakia, it is difficult to prove the application of these processes, mainly due to the insufficient monitoring of developments in the land market, including the behaviour of end-users of benefits who are operating through legal entities. It is possible to agree with several authors [32,48,73] that if the land lacks adequate legal and institutional protection, it becomes a commodity that is easily subject to manipulation and abuse. Moreover, weak governance often deprives individuals and communities of essential rights and access to land and other natural assets, and it contributes to poor land and resource management practices [73–75]. Studies from different countries have shown that in countries with low managerial efficiency and functioning of institutions, there is an increased rate of land take [76–80]. Therefore, an effective institutional framework of land-use control must be developed as one of the basic tools of sustainable land protection [77]. The state must guarantee the protection of land property rights as private goods as well as the protection of land as a public good through an effective legal and institutional framework.

In terms of the protection of property rights to agricultural land, an amendment to the Constitution of the Slovak Republic was adopted in 2017 supporting the protection of ownership of agricultural land as a non-renewable natural resource having special protection by the state and society [57]. Based on the recommendations of the European Parliament, as well as on the basis of the authorisation of the Constitution of the Slovak Republic, MARD prepared a land reform to limit the acquisition of ownership of agricultural land by entities in order to prevent speculation with land purchase for non-agricultural entities and industrial agriculture. However, land reform is still not in force because it is opposed by investors and by professional organisations, which mainly criticise the lack of relevant information to justify these restrictions. It can therefore be stated that although the state has adopted the necessary legal framework for the protection of property rights to agricultural land, it is not possible to enforce it, as the institutional framework for its implementation is absent.

Restrictions on land acquisition are highly regulated in several EU countries, especially in post-socialist countries. Typical examples of restrictions are the acquisition of land by legal entities (Hungary), the obligation to be authorised by appropriate organisations (France and Austria), permanent residence of the farmer at the place of business (Poland), etc. [29]. Additionally, the European Commission adopted the Commission Interpretative Communication on the Acquisition of Farmland and European Union Law C/2017/6168 in which it established permitted procedures for restricting the acquisition of ownership.

To the contrary, in the area of land quality protection, the state´s regulatory mechanism was proven to be malfunctioning. Research on environmental and legal liability showed that the state relies on preventive measures being ensured by the entities managing land. Although fulfilment of the measures is professionally and financially demanding, especially for small farmers, the state has not developed a methodically coordinated advisory system. Hence, it is possible to anticipate certain arbitrariness in the fulfilment of legal responsibilities. Remedy, control, and sanction mechanisms of land protection are not efficient, and relevant state bodies perform their activities at only a minimum level.

Within the context of increasing interest in land purchase with the aim of land concentration, it is the state´s malfunctioning land protection regulatory mechanism and the absence of indirect action instruments that may be key indicators leading to processes of industrial agriculture. Therefore, the adoption of legislation limiting agricultural land acquisition is important, but the processes of land grabbing presume the state´s complex provision of regulatory mechanisms and the adoption of strategic measures aimed at sustainable land quality and food security.

Based on the results, the paper provides an overview of the history and the current situation in connection with the existence of land grabbing in Slovakia. This may contribute to the scientific approach of other countries with similar conditions, mostly of Central and Eastern Europe, such as Romania, Czech Republic, Poland, Hungary, etc. Based on the results, the political and legal measures may be incorporated in the countries that emphasize the increased processes related to land grabbing. Following the contribution, further research of the specific effects of land grabbing on different groups of society (large farmers, medium-size farmers, small farmers, or other agricultural institutions and bodies, etc.) is necessary in order to reveal its positive or negative impacts.

**Author Contributions:** Conceptualisation, L.P. and Z.M.; methodology, L.P.; validation, L.P. and Z.M.; formal analysis, L.P. and Z.M.; investigation, L.P. and Z.M.; resources, L.P. and Z.M.; data curation, L.P. and Z.M.; writing—original draft preparation, L.P., A.B. and Z.M.; writing—review and editing, L.P., A.B. and Z.M..; visualisation, Z.M.; supervision, L.P. and Z.M.; project administration, L.P.; funding acquisition, L.P and A.B. All authors have read and agreed to the published version of the manuscript.

**Funding:** This research was funded by the Scientific Grant Agency of the Ministry of Education, Science, Research and Sport of the Slovak Republic (VEGA), 1/0220/18, entitled "The protection of preserving agricultural land in Slovakia".

**Institutional Review Board Statement:** Not applicable.

**Informed Consent Statement:** Informed consent was obtained from all subjects involved in the study.

**Data Availability Statement:** The authors have complied with the research and publication ethics.

**Conflicts of Interest:** The authors declare no conflict of interest. The funders had no role in the design of the study, in the collection, analyses, or interpretation of data, in the writing of the manuscript, or in the decision to publish the results.

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
