# Peer review of "Land Concentration and Land Grabbing Processes—Evidence from Slovakia"

_land, doi:10.3390/land10080873_

Round 1
Reviewer 1 Report
In my opinion, the article is relatively well written and documented both through literature references and statistical data (although the method used is very simple). However, I have a few comments:
1. Point 3 should be about the results themselves (remove the discussion from the headline - you can leave the content as it presents the results in a specific context).
2. It is worth separating the part concerning conclusions from the discussion
3. I do not know to what extent it is a matter of my software, but charts 7 and 8 fell apart - they need to be improved
4. In lines 520 to 522 there is a reference to one literature item, this should be extended to include other literature sources, additionally in lines 540 to 543 examples are given, but without references - in my opinion it should also be corrected
Author Response
Dear Reviewer,
thank you for your valuable recommendations which helped us to improve he quality of our paper.
We attached the response point-to-point down bellow.
Have a great day.
Authors

Reviewer 2 Report
The paper addresses a topic of high public and political concern and increased academic scrutiny in the critical agrarian, land use and large-scale land acquisition studies. It is a matter of analysis for diverse fields including geography and development studies. While the issue of land grabbing has mostly focused on the Global South in recent years several studies addressed land grabbing in Eastern Europe. This paper contributes to this literature.
The authors argue that large-scale land acquisitions and land concentration represent a threat for Slovakia and propose as a result of the analysis effective regulation on access to land and the protection of agricultural land. The method employed is a review of ‘historical and cultural factors’ that were conducive of land concentration and the resulting current legislative and institutional framework. The authors find that the state has adopted the necessary legal framework for the protection of property rights, however its enforcement lacks as well as the institutional environment for its implementation. Causes for this, the authors find, may be the malfunctioning land protection regulatory mechanism and absence of indirect action instruments. The authors indicate that this leads to processes of industrial agriculture. As a result, the authors recommend legislation that limits agricultural land acquisition.
Comments:
- In the abstract the authors imply that large-scale land acquisitions lead to land grabbing and that legislation needs to be enforced to stop it. The authors implicitly argue that a) land acquisitions imply land grabbing b) that they should not take place. An expectation for the reader is created that through the historical and cultural factors it is explained what land grabbing for the case of Slovakia means and how the large scale land acquisitions impact the agricultural sector and society. Through their arguments, the authors imply that land concentration has a negative impact on access to land.
A few questions arise from the abstract that I mention here for thought:
- Who does large-scale land acquisition, land grabbing and land concentration impact? The answer is rather is rather general and descriptive (small farmers and communities). What is the actual impact? Interviews with the farmers in different locations in Slovakia would shed light on how they perceive the large farms. Also, it is implied that ‘large’ and ‘small’ are homogenous groups of farmers. This is not the case and you will find different categories of farmers affected or benefiting in different ways.
- Is land grabbing equaled to large-scale land acquisition and land concentration? Analyzing the definitions of land grabbing and the actual acquisitions in place, is this really so?
- Whom do they pose a threat for? The agricultural sector, small-scale farmers, society overall? Would the limiting of agricultural land acquisitions actually not hamper smaller farms to become larger? These types of arguments need further scrutiny.
- How is large-scale land acquisition conducted? What does land concentration entail?
- How do the two limit access to land and why is agricultural land not protected? What does protection of agricultural land mean?
- The authors apply the indicated ‘method’ of historical and cultural factors analysis arguing limited data availability. Indeed, statistical data may be difficult to obtain as is common in Eastern European countries, however interviews with different agricultural enterprises, different categories of farmers, policy makers and key informants from the inputs, supply chain etc. sectors would have given an overarching insights of the matters and how these concepts (land concentration, land grabbing, access to land as well as the legislative and institutional framework behind them) are understood in the field.
- The list of references is not written in line with the journal’s requirements - see Instruction for Authors:
https://www.mdpi.com/journal/land/instructions
- The article needs the review of language conducted by a native English language speaker. See for example Lines 87-88 – dysfunctional system of public authorities; 90-91 – land grabbing occurs in particular in countries where is complicated land ownership; line 91-malfunctioning system of state bodies; line 288 – farmers for whom agriculture is a primary or secondary business conducted. The word conducted can be deleted.
- Has a copyright been received for Figure 1?
https://www.mdpi.com/authors/rights
- The introduction is quite long until the authors mention on page 4 what the aims of the paper are. Also, in this section there is no indication how the authors contribute to the broader literature on land grabbing, large-scale land acquisitions and access to land. Additionally, no theoretical framework (theory) is used to apply the historical and cultural factors as a mechanism of analysis.
- Figure 2 developed by the authors show the historical conditionalities in a vacuum. However, these develop under certain institutional contexts. The institutional theory literature can further aid here to develop the contextual background and theoretical framework. Also, the cultural factors that the authors mention to be analyzed along historical factors are missing from this figure.
- Figure three mentions data collection, however no indication to data collection is mentioned in the abstract. Authors only mention historical and cultural factors are analyzed and one thinks of a literature review here (although it is not specified how it was done). The exact methodology needs to be specified in the abstract as well as introduction. We learn only in the methods section that a survey was conducted and representatives of the Ministry of Agriculture were interviewed. Already in the abstract it needs to be mentioned that 77 sources (type of sources and how they were selected to be added) were reviewed for the literature review and 4 repres. + 104 farmers surveyed.
- According to MDPI guidelines a literature review needs to be conducted in a systematic way according to the PRISMA guidelines. My recommendation is that from the 77 sources reviewed and the land grabbing, large scale-land acquisition and access to land literature the authors devise a theoretical framework. Thus this part is not part of the methodology and methods but of the theoretical framework for the paper, which is currently lacking. The literature review needs to be formulated as such.
- Also, the statement in the abstract that because of no data availability a historical and cultural lens is applied contradicts the methods section where the authors specify that Eurostat database and the Slovak Statistical Office have been consulted.
- Lines 209-479 where Slovakia is characterized is not part of the results section but would need to be part of a section called – Study context OR Empirical context or similar.
- Figure 4 needs to be marked as Figure 4. As of now, no distinction can be made between the colours used for private and state owned land. Needs to be adjusted. Helpful if percentages are also added between brackets in the figure after private, state and unknown owners.
- When specifying the source of figures more correct to use “own representation” instead of “own processing”. Subject to the recommended language check.
- The section on the legislative framework in the results is rather descriptive. An analytical approach in conjuncture with a theoretical framework would be useful and as mentioned above for this section to be added to an Empirical Context after 1. Introduction, 2. Theoretical Framework.
- In the results section there is no understanding of what the actual results are –namely the survey conducted and the 4 interviews. How was the survey analyzed, what quantitative methods were applied to analyze the data?
- The paper is constructed on a number of assumptions that are not explained. The theoretical framework would stand as a foundation of these assumptions.
- Furthermore, certain aspects are not addressed: it is implied that large farms grab the land, however in some Eastern European countries we often find that a) small landowners are willing to sell their land as young people or remote owners are not interested to farm the land; b) certain land plots due the climatic and soil conditions are not amenable to small scale farming. If looking then at the historical factors it would be necessary to look at the agricultural structure in Slovakia even before the Soviet period. How was land structured in that period and who owned and operated the land?
- It needs to be more critically scrutinized what it meant historically for the agricultural sector that large farms have worked the land. What were the benefits? The disadvantages of land concentration and access to land can be argued by counterbalancing the information and giving an overarching picture. Setting Slovakia in the broader context of agricultural structure in Eastern European countries would also give a better insight.
- More specific comments:
- Line 28: What is meant by sustainability? Sustainability has many different definitions. It is important to understand how the authors frame it for the goal of their paper.
- Line 28: “as a part of the environment”-can be deleted, does not fit into the logic of the text
- Lines 178-179 – Authors mention several studies but only cite one. Please give a detailed overview of the studies that have revealed the gaps in terms of conceptualization, methodology and research area related to land grabbing.
- Line 268 – Slovakia belongs to the countries OR is among countries. It would be helpful if this information would be put in context – how much state owned land do other Eastern European countries have and were is Slovakia ranked among them?
- Line 304-305 – Hungary to be moved after ‘several Eu countries’
Author Response

(The authors gave the same response as above.)

Reviewer 3 Report
This article, seems to me, be correctly arranged, and clearly shows the applied method of land concentration and land grabbing processes. Figures 7 and 8 need to be corrected, at least in the pdf version we got and there is no number in Figure 4. In line 120 there is “2. Land grabbing.” and there is no “1.”. Chapter 3 is the “results and discussion” and again chapter 4 is the “discussion”. Perhaps it is better to call Chapter 4 a conclusion.
Author Response

(The authors gave the same response as above.)

Round 2
Reviewer 2 Report
Dear Authors,
Thank you for the detailed comments. My recommendations have been partly addressed and I would have a few more suggestions:
a). In the results section it is still unclear what questions the survey consisted of and how these were analyzed (what method has been used), how your analysis benefited from the additional qualitative interviews and which the results are from secondary literature. At the moment it is all presented together and there is no understanding what derives from where. You can check Bunkus and Theesfeld 2018 for reference (also regarding the structure that I had recommended in my previous comments):
Land grabbing in Europe? Socio-cultural externalities of large-scale land acquisitions in East Germany
b.) The discussion needs to be put in the broader context and address which research gaps your papers fills. What would be further research recommendations?
c). With your future responses please indicate the lines where the responses have been included in the new version. It is difficult to follow where your answers have been included.
d.) Related to the literature my recommendation was for
FELSON, M. Rural settlement and land use. Routledge, 2017. [Google Scholar]
to be cited along the reference recommendations below and include "publisher location, country, page range" in the order specified:
Author 1, A.; Author 2, B. Book Title, 3rd ed.; Publisher: Publisher Location, Country, Year; pp. 154–196.
This may also be the case with more citations: e.g. no. 45.
Author Response
Dear Reviewer,
thank you again for your response, we really appreciate your feedback and recommendations to our paper.
We incorporated all of them in resubmited manuscript.
We attach point-to point response to your review down bellow.
Have a nice day.
Authors

This manuscript is a resubmission of an earlier submission. The following is a list of the peer review reports and author responses from that submission.